# Symptoms of Depression, Anxiety and Their Co-Occurrence among Orphaned Children in Sekhukhune District, Limpopo Province

**DOI:** 10.3390/children10081279

**Published:** 2023-07-25

**Authors:** Kebogile Elizabeth Mokwena, Success Magabe, Busisiwe Ntuli

**Affiliations:** Department of Public Health, Sefako Makgatho Health Sciences University, Pretoria 0208, South Africabusisiwe.ntuli@smu.ac.za (B.N.)

**Keywords:** depression, anxiety, co-occurrence, orphans, Limpopo, Revised Child Anxiety and Depression Scale

## Abstract

Although both short- and long-term psychological challenges, specifically depression and anxiety, have been reported among orphans, there is a dearth of studies that quantify these disorders in rural settings. The aim of the study was to screen for symptoms of depression, anxiety and their co-occurrence among orphaned school-going children in rural Limpopo province, South Africa. Data were collected among primary school children in 10 schools in two villages in Limpopo province. The Revised Child Anxiety and Depression Scale (RCADS) was used to screen for symptoms of depression, anxiety and their co-occurrence among 308 orphaned learners in the selected schools. A questionnaire was used to collect socio-demographic data. STATA 13 was used to analyse the data. Descriptive statistics were used to determine the symptoms and severity of depression, anxiety and their co-occurrence. The sample of 308 consisted of the majority (60.71%) residing in Maandagshoek and being female (54.22%). Their ages ranged from 8 to 12 years, with a mean of 10.51 years. The prevalence of symptoms of depression, anxiety and co-occurrence of anxiety and depression were 23.05%, 34.09% and 32.14%, respectively. The prevalence of mental health symptoms was high among the sample. There is a need to expand the care of orphans to include mental health and not just limit their care to provide food to vulnerable children.

## 1. Introduction and Background to the Study

In the absence of parents to provide the required needs of growing children, including a sense of belonging and safety [1], orphan-hood presents gaps in the myriad of social, mental and health development among affected children. Orphan-hood also disrupts the normal development of a child, who is often compelled to take over parental responsibilities of their deceased parents, which may lead to the child feeling overwhelmed, with resultant changes in behaviour, as well as increased stress levels [2]. Orphans are children who are below the age of 18 and may be maternal (those who lost their mother), paternal (those who lost their father) or double orphans who have lost both parents due to any cause of death [3]. Although there are common challenges that affect orphans, each type has specific trends of challenges [4].

South Africa has a disproportionate number of orphans [5], with an estimated 3.7 million orphans, which was an increase of 900,000 reported two years previously [3]. Furthermore, the COVID-19 pandemic increased the number of orphans as it resulted in more deaths among adults than children. An estimated 95,000 South African children have lost at least one parent or guardian due to COVID-19 related complications [6], which increases the risk for mental health. However, as with other groups of people, mental health challenges of orphaned children are often missed; thus, they remain undiagnosed and untreated.

Orphaned children are at risk of social, health, developmental and academic challenges, as well as psychological difficulties that can manifest into psychiatric disorders [1]. Moreover, loss of parents has been associated with shorter schooling, less academic success and self-esteem challenges that can lead to children engaging in risky behaviours [2]. A common negative outcome of loss of parents is poor mental health, which includes depression, anxiety and social withdrawal [7,8].

Depression is characterised by persistent sadness and lack of interest in activities which one previously found enjoyable and can manifest as sleep disturbance, eating disorders, tiredness and poor concentration. Because sadness occurs naturally after death, it is possible to fail to differentiate between sadness and the emergence of depression among orphaned children. Anxiety, on the other hand, is manifested by nervousness, restlessness and a sense of impending danger. These symptoms result in disturbances of mood, thinking, behaviour and psychological activity. Although depression and anxiety are common mental health problems among orphaned children [9,10,11], there is a dearth of studies that quantify the prevalence of these conditions among orphaned children in South Africa and even less so in rural communities.

Factors associated with poor mental health among orphaned children include lack of counselling services, a range of medical illnesses, experiences of physical abuse and heavy responsibilities on children [8,12], as well as a lack of communication with the caregiver [12]. Orphans who had food security, access to medical services and male caregivers are reported to have better mental health [13], while the opposite is true for orphans raised in large families and who lack such basic necessities. However, most orphans in South Africa are under the care of females who are mostly unemployed and rely on social grants to meet their needs, which might make it difficult for them to have sufficient food [13]; thus, there is a greater risk of mental disorders.

The Department of Social Development is responsible for attending to the comprehensive needs of orphaned children in South Africa, with the focus being mostly on placing of the children in safe houses, which include foster care. Ideally, each child should have a social worker assigned to offer counselling services according to their needs, but because of the massive number of cases and limited number of social workers, counselling services are often not adequately accessible by children in need. However, services are often limited to being poverty-related, such as helping with obtaining identity documents, accessing grants and providing clothing and food parcels for the families [14] and less so on the mental well-being of the children, which remains largely unattended to.

The prevalence and severity of mental disorders, including the commonly occurring depression and anxiety among orphaned children in South Africa, remains largely unknown. The objective of this study was to determine the prevalence of anxiety, depression and their co-occurrence among orphaned children in two villages of Sekhukhune district of Limpopo Province, South Africa. Depression and anxiety were chosen because of their common occurrence and because there are internationally validated tools for screening them.

## 2. Methodology

The study used a quantitative and cross-sectional design, and data were collected among orphaned children in 10 primary schools in Maandagshoek and Ribacross villages in Limpopo Province. The villages are under traditional leaders and are about 30 km from the nearest town, which is Burgersfort. As with other villages in rural South Africa, the unemployment rates are high, and the majority of residents rely on small-scale farming and government social grants for survival. Most households include extended family members, and most orphans stay with their grandparents and other relatives.

The population size of 725 is composed of orphaned children who reside in and attend any of the primary schools in the two villages. A sample of convenience was conducted which takes the most accessible and available participants who were between the ages of eight and twelve and whose guardians or caregivers provided informed consent to participate in the study. Using the Raosoft sample size calculator for a population of 725, a confidence level of 95% and 5% margin of error, a minimum sample size of 252 was calculated.

Recruitment of the participants occurred from the provincial office of the Department of Basic Education, through to the district offices of the two villages, who gave permission to access the children through their schools. At the schools, the principal conducted meetings with the school governing bodies and caregivers to inform them about the study and its purpose. Information sheets were provided to caregivers as well as consent forms so that they can give consent for their children to participate in the study. At the individual level, the researcher conducted meetings with the children whose caregivers had given consent and explained the purpose of the study as well as clarifying the ethical principles governing the study. Children who were interested and willing to participate in the study were included in a list of participants to be interviewed.

### 2.1. Data Collection Tool

The 25-item Revised Child Anxiety and Depression Scale (RCADS) self-report questionnaire for child anxiety and depression was used for data collection. The tool can screen for depression, anxiety and their co-occurrence in one sitting and is reliable for cross-cultural use in a non-clinical setting, such as the school environment [15], as was needed in this study. It has sufficient structural validity, internal consistency, test–retest reliability and criterion validity [16] and was also previously used in South Africa [17].

### 2.2. Data Collection

A pilot study preceded main data collection. Data were collected by the researcher and a trained research assistant. On the day of data collection, the participants who qualified to participate in the study and whose parents had provided written informed consent were assembled in a classroom. The purpose of the study was explained, and they were given an opportunity to ask questions or seek clarification, and when all questions were answered, they were requested to sign the assent form. Participants who were aged 8 to 10 were individually interviewed by the researcher and her assistant who asked questions and filled the questionnaires. Each of the participants aged 11–12 years old had a table which enabled self-administration of the questionnaire as the researcher read out the questions for them to tick the answers in their questionnaires. The data collection took an estimated 45 min, and no incentives were given to the participants.

### 2.3. Ethical Considerations

Ethical clearance was obtained from the Sefako Makgatho Health Sciences University Research Ethics Committee (SMUREC). Permission to conduct the study at the schools was granted by the Limpopo Provincial Department of Education, Limpopo Provincial Research Ethics Committee, the district and circuit managers and the principals and school governing bodies of the ten schools. Written informed consent was provided by the primary caregivers of the children, and the children provided assent to participate in the study.

### 2.4. Data Analysis

A total 308 orphaned children participated in the study. Socio-demographic data were descriptively analysed using STATA 13 software. The Revised Child Anxiety and Depression Scale (RCADS version 3.0), which is an automatic scoring excel sheet, was used to quantify depression, anxiety and their co-occurrence. The raw data from the questionnaires were entered into the RCADS-25 program to acquire T-scores for total anxiety, total depression and their total co-occurrence, which were automatically produced by the program after entering the participants’ scores. Participants with T-scores of less than 65, which are scores below the borderline clinical threshold, were categorized as having no symptoms of depression, anxiety and their co-occurrence. Scores between 65 and 69, which are at the borderline clinical threshold level, were categorized as having mild symptoms, and scores of above 70 were classified as severe symptoms [18]. The total T scores were therefore classified into three categories and coded as 0 = no symptoms, 1 = mild symptoms and 2 = severe symptoms and then entered in STATA 13 software program as outcome variables during analysis.

To explore for associations between the outcome variables, i.e., anxiety symptoms, depression symptoms and their co-occurrence, and a range of socio-demographic variables, a chi square test of association was conducted for each outcome variable. The anxiety scores were converted to categories of anxious and not; depression scores were converted to categories of depressed and not; and their co-occurrence scores were converted to co-occur and not.

## 3. Results

### 3.1. Socio-Demographic Characteristics of the Sample

The majority lived in Maandagshoek (60.71%, n = 187), were female (54.22%, n = 167), were in grade 5 or lower (55.84%, n = 172) and had no one employed in their household (73.7%, n = 227). Table 1 below shows the rest of the socio-demographics of the participants.

### 3.2. The Prevalence of Depression Symptoms

The prevalence of depression symptoms, i.e., scores of 65 and above, was 23.05% (n = 71), with the proportion of those with severe symptoms, (scores above 70) more than double of those with mild symptoms (scores between 65 and 69). Of those who tested positive for depression, 67.6% (n = 48) had severe symptoms, compared to 32.4% (n = 23) who had mild symptoms. Figure 1 below shows the prevalence and severity categories of the T scores for depression symptoms.

### 3.3. Prevalence of Anxiety Symptoms

Figure 2 below shows the prevalence and severity of anxiety symptoms.

### 3.4. Co-Occurrence of Anxiety and Depression Symptoms

Figure 3 below shows the prevalence and severity of co-occurring of depression and anxiety.

The Pearson Chi Square test was conducted to explore associations between anxiety symptoms, depression symptoms and their co-occurrence and socio-demographic variables, using a *p*-value of 0.05. Table 2 below shows the results of significant associations.

## 4. Discussion

The almost equal proportion of maternal and paternal orphans (41.23% and 41.88%) is contradictory to a national South African study which reported that prevalence of paternal orphan-hood was higher, followed by double orphan-hood and then maternal orphan-hood [19]. The high proportion of paternal orphans was explained by the higher death rates among men, with violence as a significant contributory factor [4]. The higher proportion of girls is similar to the results of a previous study [5], but that study reported a much lower proportion of double orphans at 6%, which is more than double in the current study.

The majority (78.57%) of the caregivers were female, which supports previous findings that the majority of orphans are cared for by females [20,21,22]. On the other hand, females are reported to be financially disadvantaged when compared to men, which highlights the socio-economic difficulties of the children under their care. Such socio-economic difficulties further increase mental health challenges among the affected children [23,24], with potential long-term consequences.

Despite the fact that 41.21% of the participants in the current study are maternal orphans and their fathers are still alive, only 8.12% of them reported to be under the care of their fathers, which suggests that paternal orphans are more likely to remain under the care of their mothers while maternal orphans are less likely to remain under the care of their fathers. A significant difference from other studies on orphans is that none of the participants reported living in child-headed households, which resonates with previous findings [19], who reported very low cases of child headed households in Limpopo. As with other communities in rural South Africa, fewer than a quarter of the participants had caregivers who were employed, which suggests economic difficulties which further contribute to mental health challenges for the households, including the orphans in those households.

The mental challenges among the sample the sample is concerning, especially in the context of the high number of orphans in the country. The assistance to orphans is generally focussed on and limited to providing them with shelter and food, and not much attention is paid to their ongoing mental health needs. This may explain the ongoing negative outcomes regarding schooling, self-esteem and other social measurements of the affected orphans. The results of this study highlight essential gaps in the provision of services for orphans, which, if attended to, may make a difference to the outcomes of orphans.

The 23% prevalence of depression symptoms was similar to the prevalence of 24.1% in Ethiopia [8] and the 20.8% prevalence in Nepal [25], which suggests that depressive symptoms among the current sample of orphans fall within the norm. However, the prevalence was slightly higher than the 21% reported in Tshwane [26]. The differences may be due to the differences in social support between urban and rural South Africa, the use of different screening tools and differences in the age ranges of the samples.

Orphan-hood is a major traumatic event which has been associated with a range of social risks, including low rates of school attendance, hunger, child labour and vulnerability to sexual abuse among girls [27]. The mental health outcomes of such exposure to undesirable conditions will be determined by a range of conditions, which includes socio-demographic, support provided and the personality traits of the affect child.

The prevalence of anxiety was 34.09%, with the proportion of those experiencing severe symptoms being twice of those with mild symptoms. The results of the current study are supported by previous studies that confirm high prevalence of anxiety among orphans. In a similar study conducted in conflict-riddled Gaza strip, a comparable prevalence of 31% was reported [11], which puts into perspective the proportion of orphans in South Africa whose prevalence of anxiety is comparable to those experienced by orphans in areas of conflict.

Of concern is that anxiety symptoms in children have been associated with other psychiatric disorders [28], which are likely to be missed as orphans have general lack of access to health services, with serious consequences, as such problems often persist among orphaned children [29,30]. The results also highlight the need to address mental health matters in the country, from measurement of the problems, to providing services and measures for prevention.

The co-occurrence of depression and anxiety is common among both adults and children [31]. In children, the co-occurrence has been associated with a range of health and social problems, which include increased risk of co-occurring conditions, poor use of health care, school problems and high parenting aggravation [32], which may have long-term impacts. The current study found that the prevalence of co-occurring anxiety and depression was 32.14%, with those with severe symptoms being more than those with mild symptoms. These results highlight a situation in which the proportion of children who experience severe symptoms of both anxiety and depression is higher than those who experience mild symptoms.

In South Africa, there is lack of adequate studies which measures the co-occurrence of symptoms of anxiety and depression, as most researchers focus on studying depression and anxiety independently [9,11,33]. While this is still valuable, such studies may have resulted in forming an incomplete picture of the mental status of the participants; hence, there are advantages of using the Revised Child Anxiety and Depression Scale, which measures depression and anxiety separately but also their co-occurrence. From the results of their studies, the screening tools that they used probably screened the two mental disorders separately, or they used a different tool for each disorder.

The implications of these findings are significant, because losing a parent in childhood is a major traumatic experience, and such childhood traumatic experiences have both short- and long-term impacts on the child’s well-being, such as increased risk for the development of schizophrenia, whose symptoms persist throughout adult life for most patients [34]. Other consequences of childhood trauma include increased risk for adult mental disorders such as depression, impaired working memory, compromised executive function and psychosis [35].

Moreover, childhood trauma is a risk factor for a range of psychopathology across development, which implies that the affected children’s development may be impacted negatively at social and emotional levels, which requires early interventions to assist in the prevention of emergence of psychopathology among these children [36]. With the current reported increase in global mental disorders, this group of children require long-term attention and care, if they are to be assisted to thrive in future.

The prevalence of mental disorders among children, including depression and anxiety and their co-occurrence, are influenced by a myriad of factors, which may be grouped as environmental, social and even personal. However, suggestions for considerations of living arrangements, such as whether the child lives with other siblings, and caregiver attachments, which is challenging to measure, have been made [37]. The current study focussed on quantifying the symptoms and on the contextual factors.

The socio-demographic factors associated with the three outcome variables, i.e., symptoms of anxiety, depression and their co-occurrence, suggests that orphaned children need customized assistance packages, depending on their age (and correlated school grade), gender and living arrangements for all three outcomes. This highlights the need for additional resources to meet the needs of orphaned children, as neglecting to provide such support at the opportune time may have lasting negative outcomes for their future live.

The other associated factors, i.e., the number of children in the household, the placing of the child in foster care and who the head of the household is, are factors that cannot be modified or changed to make it better for the affected child. However, these can be addressed by implementing interventions custom made for the specific situation.

### 4.1. Future Research Implications

The occurrence of mental health challenges is associated with, among others, age and grade. This suggests the need for age-related mental health interventions for orphaned children.

### 4.2. Practical Clinical Implications

There is a need for regular assessment of the mental status of children who are at risk of mental disorders, especially because there are easy to use tools, such as the Revised Child Anxiety and Depression Scale, which can also be used to assess any changes in mental status over time.

### 4.3. Limitations of the Study

As with other cross-sectional studies, the results are limited to a snapshot of the mental status of the participants during data collection, which does not enable the studying of mental status over a period of time.

### 4.4. Strengths of the Study

A major strength of the study was the use of RCADS screening tool which can screen for depression, anxiety and their co-occurrence in one sitting. This also enables comprehensive assessment of a child without compartmentalizing his or her mental status, as each of the components affect the others.

## 5. Conclusions

The prevalence of depression, anxiety and their co-occurrence among orphaned children in the current study is much higher than the 20% estimated among children in both developed and developing countries [38]. The prevalence of severe symptoms of both depression and anxiety was twice that of mild symptoms, which highlights the need to consider the mental health care needs of orphaned children, which should include measurement, prevention and management. This study further highlights the importance of addressing the shortage of mental health services, particularly among children, which represent up to 50% of some populations [39]. Lastly, there is a need to allocate resources for a long-term follow-up system for orphaned children, to attend to their varied needs.

## 6. Recommendations

The following recommendations emanate from this study:The Department of Social Development, which is the government entity responsible for the well-being of vulnerable children, should formally integrate routine mental health assessment and service provision for all orphaned children.Further study attention be given to causes of orphan-hood and responsive interventions, as there are reports that the severity of mental challenges may be influenced by the cause of the parents’ deaths [40,41,42]. This may therefore enable the development of custom-made mental health interventions for orphans.In the context of the many orphans of school-going ages, schools be provided with resources to develop programs to screen for mental health challenges among children, to enable referral to appropriate services.

## Figures and Tables

**Figure 1 children-10-01279-f001:**
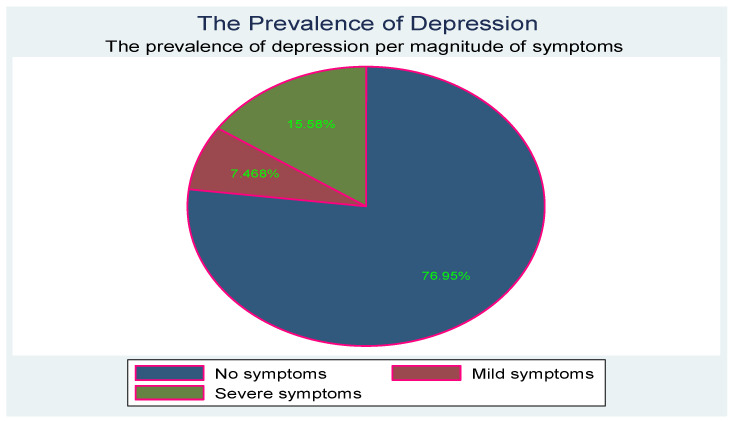
Prevalence and severity of depression symptoms.

**Figure 2 children-10-01279-f002:**
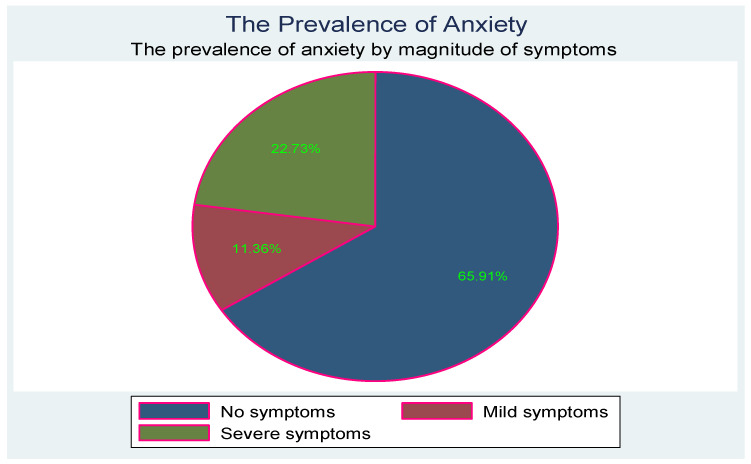
Prevalence and severity of anxiety symptoms.

**Figure 3 children-10-01279-f003:**
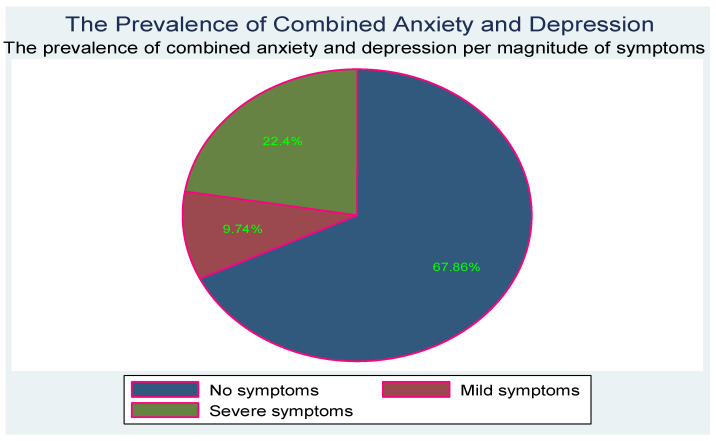
Co-occurrence of depression and anxiety symptoms.

**Table 1 children-10-01279-t001:** Socio-demographic characteristics of participants.

Variables		Frequency	Percentage
Total Number of Participants		308	100
Village	Maandagshoek	187	60.71
	Ribacross	121	39.29
Gender of participants	Females	167	54.22
	Males	141	45.78
Age of participants	8–10 years	132	42.86
	11–12 years	176	57.14
Grade	Up to 5	172	55.84
	6 and 7	136	44.16
Deceased parent/s	Both parents	52	16.88
	Father only	129	41.88
	Mother only	127	41.23
Caregiver	Aunt	31	10.06
	Brother	2	0.65
	Father	25	8.12
	Grandfather	1	0.32
	Grandmother	109	35.39
	Mother	115	37.34
	Sister	18	5.84
	Uncle	7	2.27
Age category of caregiver	Adult	173	56.17
	Older person	91	29.55
	Youth	44	14.29
Gender of caregiver	Female	273	88.64
	Male	35	11.36
Employment status of caregiver	Employed	74	24.03
	Unemployed	234	75.97
Head of household	Grandparent	145	47.08
	Parent	123	39.94
	Sibling	12	3.90
	Other relatives	28	9.09
Number of people in household	0 to 5	107	34.74
	6 and above	201	65.26
Number of siblings	0 to 3	199	64.61
	4 and above	109	35.39
Number of children in household	0 to 4	229	74.35
	5 and above	79	25.65
Number of people employed in household	None	227	73.70
	1 to 2	59	19.16
	3 and above	22	7.14

**Table 2 children-10-01279-t002:** Associations between symptoms of anxiety, depression and their co-occurrence and socio-demographic variables.

Variable	*p*-Value
Anxiety symptoms	
Age	0.001
Gender	0.004
Number of children in household	0.002
Placed in foster care	0.000
Head of household	0.002
Depression symptoms	
Grade	0.005
Number of children in household	0.004
Placed in foster care	0.004
Enough food	0.033
Co-occurrence of anxiety and depression symptoms	
Grade	0.046
Placed in foster care	0.001

## Data Availability

Data availability is guided by data sharing principles of Sefako Makgatho Health Sciences University.

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
