# Peer review of "Symptoms of Depression, Anxiety and Their Co-Occurrence among Orphaned Children in Sekhukhune District, Limpopo Province"

_children, 2023, doi:10.3390/children10081279_

Round 1

Reviewer 1 Report

Dear Authors,

your paper addresses important issues related to mental health of orphans in South Africa. It is particularly interesting that you study rural children.

Bu there are some major concerns about the paper: to me it looks like a purely sociologic study that just reports levels of depression and anxiety with no attempt of analysis. Overall, there is no analysis other than "there were that many orphans with depressive symptoms". Obviously, even wothin data you have you could dig deeper in your analysis.

The text needs moderate English editing. Some times it is really hard to understand the idea you wanted to express.

Author Response

The response to the reviewers are attached 

Reviewer 2 Report

Abstract

- To avoid repeating the same information: “in 10 schools in two villages in Limpopo province” and “from 10 schools in the 2 villages”.

Introduction

- Overall, the Introduction provides the literature review about the topic and contextualizes the goals of the study. However, some of the contents could be better organised and contextualized.

- They could better explain why they chose to study depression and anxiety in orphans, compared to other health or psychosocial problems they also experience.

- The prevalence of depression and anxiety and their co-occurrence may vary according to other environmental, family, and social factors that have not been considered. What explanatory models were used in this study?

- Other empirical studies with similar samples in other geographical areas could have been explored to compare the results.

Minor point

-   Page 2, third paragraph: remove the comma before the reference [1].

Methods

- The description of the Method is compartmentalized into too many subtopics. It is recommended to reorganize the information according to the more standardized structure of scientific articles.

Results

- All analysis of results requires review, both from a technical point of view and their presentation.

Discussion

- The discussion includes interesting aspects. However, after reviewing the Introduction, it is recommended to further adjust the Introduction, Results and Discussion.

References

- All references should be reviewed. Some are incomplete or incorrectly formatted.

- Whenever possible, the authors should add the DOI.

Minor editing of English language required.

Author Response

the response is attached

Reviewer 3 Report

Dear Authors,

Congratulations on your interesting work, concerning Symptoms of depression, anxiety and their co-occurrence among orphaned children in Sekhukhune district, Limpopo Province

I suggest some minor revisions:

Results

Please add a patient flowchart  (a figure).

Analysis of relationships between sociodemographic data and screening tools (depression, anxiety) is missing.

Analysis of prevalence of anxiety and depression symptoms only seems to be too poor, too concise.

Discussion

Results of analysis of relationships between sociodemographic data and screening tools  should be discussed.

Please reformulate Discussion section. The conclusions should follow the  future research implications, practical clinical implications, study limitations, strenghts of the study.

Author Response

The response is attached

Round 2

Reviewer 1 Report

Thank you  for your clarifications and added discussion

Reviewer 3 Report

The answer form for revierers is empty (however revisions have been made)